# Anti-Spoofing Method for Improving GNSS Security by Jointly Monitoring Pseudo-Range Difference and Pseudo-Range Sum Sequence Linearity

**DOI:** 10.3390/s23208418

**Published:** 2023-10-12

**Authors:** Xinran Zhang, Taotao Liang, Jie Tian, Junwei Wu, Chuan Wang, Maolin Chen

**Affiliations:** Institute of Electronic Engineering, China Academy of Engineering Physics, Mianyang 621999, China; zhang-xr15@tsinghua.org.cn (X.Z.); oldphoto522121@163.com (T.L.); tianjie@caep.cn (J.T.); wjwcaep@163.com (J.W.); wch_eml@126.com (C.W.)

**Keywords:** Global Navigation Satellite System (GNSS), security, spoofing interference, pseudo-range difference (PRD), pseudo-range sum (PRS), linearity, sum of squares of errors (SSE)

## Abstract

Spoofing interference is one of the most emerging threats to the Global Navigation Satellite System (GNSS); therefore, the research on anti-spoofing technology is of great significance to improving the security of GNSS. For single spoofing source interference, all the spoofing signals are broadcast from the same antenna. When the receiver is in motion, the pseudo-range of spoofing signals changes nonlinearly, while the difference between any two pseudo-ranges changes linearly. Authentic signals do not have this characteristic. On this basis, an anti-spoofing method is proposed by jointly monitoring the linearity of the pseudo-range difference (PRD) sequence and pseudo-range sum (PRS) sequence, which transforms the spoofing detection problem into the sequence linearity detection problem. In this paper, the model of PRD and PRS is derived, the hypothesis based on the linearity of PRD sequence and PRS sequence is given, and the detection performance of the method is evaluated. This method uses the sum of squares of errors (SSE) of linear fitting of the PRD sequence and PRS sequence to construct detection statistics, and has low computational complexity. Simulation results show that this method can effectively detect spoofing interference and distinguish spoofing signals from authentic signals.

## 1. Introduction

Global Navigation Satellite System (GNSS) has the unique technical characteristics of all-weather, all-day, and global coverage and has become the most widely used space-based radio positioning system at present [1,2]. While it is widely used, the security of GNSS has been paid more attention [3]. Due to the inherent vulnerability of GNSS, such as weak signal power, publicly known civil signal structure, slow message update, etc., GNSS is vulnerable to external interference, and its security is seriously threatened [4,5]. In its current form, the main types of GNSS interference include suppression interference and spoofing interference. Suppression interference interferes with the target receiver by transmitting high-power signals, which reduces its positioning performance and even makes it impossible to locate [6]. However, the effect of suppressing interference is dominant. It is easy for the attacked receiver to find out that it has been suppressed and take countermeasures accordingly. The more covert spoofing interference induces the receiver to output wrong positioning results by broadcasting false signals with the same structural characteristics and different parameters as GNSS signals [7]. Compared with suppression interference, spoofing interference does not require high power, and its interference effect is more concealed, so its potential harm is higher [8]. The research on anti-spoofing technology is of great significance to improving the security of GNSS, which has gradually been widely valued by academic circles and has become a new research hotspot [9,10].

According to the number of spoofing sources, spoofing interference can be divided into two modes: single spoofing source interference and multiple spoofing sources interference [11]. Single spoofing source interference means that a spoofing source transmits multiple spoofing signals, which is the most common spoofing interference mode at present. In this way, the spoofing signals can be consistent without knowing the target receiver position [12]. Multiple spoofing sources interference means that each spoofing source produces a spoofing signal, and all the spoofing signals are transmitted to the receiver from different directions. This way is closer to the actual scene. Nevertheless, to make the spoofing signals meet the consistency, the receiver position should be known, and the spoofing sources need to synchronize accurately through the communication link, which requires high implementation cost and technical level. Up to now, the interference mode of multiple spoofing sources cooperative spoofing is still in the theoretical research stage, and there is no public report in the literature.

Because of the complexity of spoofing interference, the principles of anti-spoofing methods are quite different. The anti-spoofing method based on *C*/*N*0 monitoring assumes that the *C*/*N*0 of spoofing signals exceeds the normal range, but if the spoofing signals are broadcast together with noise, this method will be invalidated [13]. The signal quality monitoring method can detect the traction spoofing interference based on the fact that overlapping correlation peaks in the traction process will lead to distortion of the autocorrelation function but is ineffective when correlation peaks are separated [14]. The receiver autonomous integrity monitoring method detects spoofing interference according to the consistency of authentic signals, but it will fail when spoofing signals tend to be consistent [15].

For single spoofing source interference, an important feature is that all spoofing signals are broadcast from the same antenna. Therefore, the incidence angles of all spoofing signals are the same, while those of authentic signals are different. According to this, methods based on multiple antennas or multiple receivers can effectively detect spoofing interference [16,17,18,19,20,21]. However, these methods not only need to synchronize the observations of all antennas or receivers but also need to add extra hardware, which leads to high complexity and cost. In addition, methods based on mobile receivers are also effective [22,23,24]. All spoofing signals have the same propagation path, while the authentic signals from different satellites have different propagation paths. Therefore, when the receiver moves, some observations of the authentic signals and the spoofing signals will show different changes. Previous studies assumed that the channel gains or carrier Doppler shifts of all the spoofing signals are correlated, so spoofing interference can be detected by monitoring the correlation of channel gains or carrier Doppler shifts between different signals. However, the change in channel gain is related to the distance between the spoofing source and the receiver. The farther the distance, the less obvious the channel gain variation caused by the receiver motion [11]. In addition, the change in carrier Doppler shift is related to the velocity change in the receiver. The smaller the velocity change, the less obvious the carrier Doppler shift variation caused by the receiver motion.

Based on the above considerations, this paper proposes an anti-spoofing method for single spoofing source interference, which jointly monitors the linearity of the pseudo-range difference (PRD) sequence and pseudo-range sum (PRS) sequence. In a short time, the pseudo-range variation caused by the satellite motion is linear. For spoofing signals, although the variations in pseudo-ranges caused by the receiver motion are nonlinear, these variations are the same. Consequently, the pseudo-range of spoofing signals changes nonlinearly, while the difference between any two pseudo-ranges changes linearly. For authentic signals, it does not have this characteristic. Therefore, we can detect spoofing interference by monitoring the linearity of the PRD sequence and PRS sequence and distinguishing between authentic signals and spoofing signals. Compared with the existing research, the proposed method can effectively solve the problem of spoofing detection even when the distance between the spoofing source and the receiver is far away or the velocity of the receiver changes slowly, which shows that the performance of the method is more robust.

Subsequent sections of this paper are arranged as follows. In Section 2, the models of PRD and PRS are derived, and their linear problems are analyzed. In Section 3, hypothesis testing is given, linearity detection statistics are constructed, and false alarm probability and detection probability are analyzed. Section 4 shows the simulation results. Section 5 summarizes the content of this paper.

## 2. Linearity Analysis

### 2.1. Pseudo-Range Sequence Linearity Analysis

The pseudo-range measurement value *ρ* of the authentic signal is as shown in Equation (1):(1)$$\rho =r+c\left(\delta t_{u}-\delta t_{s}+I+T\right)+\varepsilon ,$$
where r represents the geometric distance between satellite and receiver; c represents the speed of light; $\delta t_{u}$ and $\delta t_{s}$ represent receiver clock error and satellite clock error, respectively; I is ionospheric delay; T is tropospheric delay; and ε represents pseudo-range measurement noise, which can be considered as Gaussian white noise.

Define the pseudo-range sequence ${\vec{\rho }}_{i}=\left[\begin{array}{ccc} \rho_{i}\left(0\right), & \begin{array}{cc} \cdots & \rho_{i}\left(k\right), \end{array} & \begin{array}{cc} \cdots & \rho_{i}\left(K-1\right) \end{array} \end{array}\right]$. $\rho_{i}\left(k\right)$ denotes the pseudo-range measurement value of the signal *i* at the k-th measurement time. The measurement time interval is Δt, and the sequence length is K. In a short time, the changes in ionospheric delay, tropospheric delay, and satellite clock error are very small and can be ignored. In addition, since the distance between the satellite and receiver is far enough, the position change of the satellite and receiver in a short time is very small relative to the distance between them. Therefore, the unit observation vector ${\vec{l}}_{i}$ of the satellite at the receiver can be considered constant. Thus, $\rho_{i}\left(k\right)$ can be approximately expressed as
(2)$$\rho_{i}\left(k\right)\approx \Delta {\vec{r}}_{s,i}\left(k\right)\cdot {\vec{l}}_{i}-\Delta {\vec{r}}_{u}\left(k\right)\cdot {\vec{l}}_{i}+c\cdot \delta t_{u}\left(k\right)+\varepsilon_{i}\left(k\right)+{\mathrm{Cn}}_{i}.$$

In Equation (2), ${\mathrm{Cn}}_{i}=\rho_{i}\left(0\right)-c\cdot \delta t_{u}\left(0\right)-\varepsilon_{i}\left(0\right)$ is a constant; $\Delta {\vec{r}}_{s,i}\left(k\right)$ and $\Delta {\vec{r}}_{u}\left(k\right)$ represent the position vectors of the satellite and the receiver from time 0 to time k, respectively.

$\Delta {\vec{r}}_{s,i}\left(k\right)\cdot {\vec{l}}_{i}$ means the radial projection component of the satellite position change. The radial acceleration of the satellite is less than $2\times 10^{-5}\;m/s^{3}$ [11]. The radial acceleration is so small that it has little effect on radial velocity. Therefore, in a short time, it can be considered that the radial velocity is constant, which means that $\Delta {\vec{r}}_{s,i}\left(k\right)\cdot {\vec{l}}_{i}$ varies linearly. The receiver is driven by a stable clock, and its clock drift does not change obviously with time [25]. Therefore, the receiver clock difference $\delta t_{u}\left(k\right)$ can also be considered to vary linearly in a short time. When the receiver velocity is constant, $\Delta {\vec{r}}_{u}\left(k\right)\cdot {\vec{l}}_{i}$ changes linearly with time. When its velocity changes, $\Delta {\vec{r}}_{u}\left(k\right)\cdot {\vec{l}}_{i}$ changes nonlinearly. According to the above analysis, the elements in ${\vec{\rho }}_{i}$ are linear when the receiver moves at a uniform velocity, and nonlinear when the receiver moves at a variable velocity.

In case of spoofing signal, under the assumption that the position of the spoofing source is unchanged, $\rho_{i}\left(k\right)$ can be approximately expressed as
(3)$$\rho_{i}\left(k\right)\approx \Delta {\vec{r}}_{s,i}\left(k\right)\cdot {\vec{l}}_{i}^{p}-\Delta {\vec{r}}_{u}\left(k\right)\cdot {\vec{l}}^{u}\left(k\right)+\Delta \rho_{p,i}\left(k\right)+c\cdot \delta t_{u}\left(k\right)+\varepsilon_{i}\left(k\right)+{\mathrm{Cn}}_{i},$$
where ${\vec{l}}_{i}^{p}$ represents the unit observation vector of the satellite at the spoofing source, ${\vec{l}}^{u}\left(k\right)$ represents the unit observation vector of the spoofing source at the receiver, and $\Delta \rho_{p,i}\left(k\right)$ represents the difference between the additional component of the spoofing source at time k and time 0. It should be emphasized that in addition to the assumption that the spoofing source is stationary, we also need to assume that the additional component of the spoofing source to any signal changes linearly, which is the prerequisite for the effectiveness of the proposed method. Thus, $\Delta \rho_{p,i}\left(k\right)$ can be considered to vary linearly.

Since the distance between the satellite and the spoofing source is far enough, ${\vec{l}}_{i}^{p}$ can be considered to be constant in a short time, so that $\Delta {\vec{r}}_{s,i}\left(k\right)\cdot {\vec{l}}_{i}^{p}$ varies linearly with time. While for ${\vec{l}}^{u}\left(k\right)$, due to the distance between the spoofing source and the receiver being relatively close, ${\vec{l}}^{u}\left(k\right)$ will change continuously with the movement of the receiver. Therefore, whether the receiver is moving at a uniform velocity or not, $\Delta {\vec{r}}_{u}\left(k\right)\cdot {\vec{l}}^{u}\left(k\right)$ is nonlinear with time, and then the elements in ${\vec{\rho }}_{i}$ are nonlinear.

### 2.2. PRD Sequence Linearity Analysis

Defining $\Delta {\vec{\rho }}_{ij}=\left[\begin{array}{ccc} \Delta \rho_{ij}\left(0\right), & \begin{array}{cc} \cdots & \Delta \rho_{ij}\left(k\right), \end{array} & \begin{array}{cc} \cdots & \Delta \rho_{ij}\left(K-1\right) \end{array} \end{array}\right]$ is the sequence of measured pseudo-range differences between the signal i and the signal j, where $\Delta \rho_{ij}\left(k\right)$ can be denoted as
(4)$$\Delta \rho_{ij}\left(k\right)=\rho_{i}\left(k\right)-\rho_{j}\left(k\right).$$

If the two signals are authentic signals, $\Delta \rho_{ij}\left(k\right)$ can be derived as
(5)$$\Delta \rho_{ij}\left(k\right)\approx \Delta {\vec{r}}_{s,i}\left(k\right)\cdot {\vec{l}}_{i}-\Delta {\vec{r}}_{s,j}\left(k\right)\cdot {\vec{l}}_{j}-\Delta {\vec{r}}_{u}\left(k\right)\cdot \left({\vec{l}}_{i}-{\vec{l}}_{j}\right)+\varepsilon_{i}\left(k\right)-\varepsilon_{j}\left(k\right)+\Delta {\mathrm{Cn}}_{ij},$$
where $\Delta {\mathrm{Cn}}_{ij}=\left(\rho_{i}\left(0\right)-\varepsilon_{i}\left(0\right)\right)-\left(\rho_{j}\left(0\right)-\varepsilon_{j}\left(0\right)\right)$ is a constant. The pseudo-range variation $c\cdot \delta t_{u}\left(k\right)$ caused by the change in the receiver clock difference is eliminated in the calculation process. In a short time, $\Delta {\vec{r}}_{s,i}\left(k\right)\cdot {\vec{l}}_{i}$ and $\Delta {\vec{r}}_{s,j}\left(k\right)\cdot {\vec{l}}_{j}$ change linearly, and ${\vec{l}}_{i}-{\vec{l}}_{j}$ can be considered unchanged. Therefore, it is not difficult to conclude that the elements in $\Delta {\vec{\rho }}_{ij}$ are linear when the receiver moves at a uniform velocity and nonlinear when the receiver moves at a variable velocity.

If the two signals are spoofing signals, $\Delta \rho_{ij}\left(k\right)$ can be derived as
(6)$$\Delta \rho_{ij}\left(k\right)\approx \Delta {\vec{r}}_{s,i}\left(k\right)\cdot {\vec{l}}_{i}^{p}-\Delta {\vec{r}}_{s,j}\left(k\right)\cdot {\vec{l}}_{j}^{p}+\Delta \rho_{p,i}\left(k\right)-\Delta \rho_{p,j}\left(k\right)+\varepsilon_{i}\left(k\right)-\varepsilon_{j}\left(k\right)+\Delta {\mathrm{Cn}}_{ij}.$$

The pseudo-range change $\Delta {\vec{r}}_{u}\left(k\right)\cdot {\vec{l}}^{u}\left(k\right)$ caused by the change in receiver position is eliminated in the calculation process. In this case, the elements in $\Delta {\vec{\rho }}_{ij}$ are linear regardless of whether the receiver is moving at a uniform velocity or a variable velocity.

If one of the two signals is an authentic signal and the other is a spoofing signal, $\Delta \rho_{ij}\left(k\right)$ can be derived as
(7)$$\begin{array}{c} \Delta \rho_{ij}\left(k\right)\approx \Delta {\vec{r}}_{s,i}\left(k\right)\cdot {\vec{l}}_{i}-\Delta {\vec{r}}_{s,j}\left(k\right)\cdot {\vec{l}}_{j}^{p}-\Delta {\vec{r}}_{u}\left(k\right)\cdot \left({\vec{l}}_{i}-{\vec{l}}^{u}\left(k\right)\right)\\ -\Delta \rho_{p,j}\left(k\right)+\varepsilon_{i}\left(k\right)-\varepsilon_{j}\left(k\right)+\Delta {\mathrm{Cn}}_{ij} \end{array}.$$

Since ${\vec{l}}^{u}\left(k\right)$ will change during the motion of the receiver, $\Delta {\vec{r}}_{u}\left(k\right)\cdot {\vec{l}}^{u}\left(k\right)$ changes nonlinearly with time regardless of whether the receiver moves at a uniform velocity, which means that the elements in $\Delta {\vec{\rho }}_{ij}$ are nonlinear.

### 2.3. PRS Sequence Linearity Analysis

Defining $\Sigma {\vec{\rho }}_{ij}=\left[\begin{array}{ccc} \Sigma \rho_{ij}\left(0\right), & \begin{array}{cc} \cdots & \Sigma \rho_{ij}\left(k\right), \end{array} & \begin{array}{cc} \cdots & \Sigma \rho_{ij}\left(K-1\right) \end{array} \end{array}\right]$ is the sequence of measured pseudo-range sums between the signal i and the signal j, where $\Sigma \rho_{ij}\left(k\right)$ can be denoted as
(8)$$\Sigma \rho_{ij}\left(k\right)=\rho_{i}\left(k\right)+\rho_{j}\left(k\right).$$

If the two signals are authentic signals, $\Sigma \rho_{ij}\left(k\right)$ can be derived as
(9)$$\begin{array}{c} \Sigma \rho_{ij}\left(k\right)\approx \Delta {\vec{r}}_{s,i}\left(k\right)\cdot {\vec{l}}_{i}+\Delta {\vec{r}}_{s,j}\left(k\right)\cdot {\vec{l}}_{j}-\Delta {\vec{r}}_{u}\left(k\right)\cdot \left({\vec{l}}_{i}+{\vec{l}}_{j}\right)+2c\cdot \delta t_{u}\left(k\right)\\ +\varepsilon_{i}\left(k\right)+\varepsilon_{j}\left(k\right)+\Sigma {\mathrm{Cn}}_{ij} \end{array},$$
where $\Sigma {\mathrm{Cn}}_{ij}=\left(\rho_{i}\left(0\right)-c\cdot \delta t_{u}\left(0\right)-\varepsilon_{i}\left(0\right)\right)+\left(\rho_{j}\left(0\right)-c\cdot \delta t_{u}\left(0\right)-\varepsilon_{j}\left(0\right)\right)$ is a constant. In a short time, $\Delta {\vec{r}}_{s,i}\left(k\right)\cdot {\vec{l}}_{i}$ and $\Delta {\vec{r}}_{s,j}\left(k\right)\cdot {\vec{l}}_{j}$ change linearly, and ${\vec{l}}_{i}+{\vec{l}}_{j}$ can be considered unchanged. Therefore, it is not difficult to conclude that the elements in $\Sigma {\vec{\rho }}_{ij}$ are linear when the receiver moves at a uniform velocity and nonlinear when the receiver moves at a variable velocity.

If the two signals are spoofing signals, $\Sigma \rho_{ij}\left(k\right)$ can be derived as
(10)$$\begin{array}{c} \Sigma \rho_{ij}\left(k\right)\approx \Delta {\vec{r}}_{s,i}\left(k\right)\cdot {\vec{l}}_{i}^{p}+\Delta {\vec{r}}_{s,j}\left(k\right)\cdot {\vec{l}}_{j}^{p}-2\Delta {\vec{r}}_{u}\left(k\right)\cdot {\vec{l}}^{u}\left(k\right)+\Delta \rho_{p,i}\left(k\right)+\Delta \rho_{p,j}\left(k\right)\\ +2c\cdot \delta t_{u}\left(k\right)+\varepsilon_{i}\left(k\right)+\varepsilon_{j}\left(k\right)+\Sigma {\mathrm{Cn}}_{ij} \end{array}.$$

If one of the two signals is an authentic signal and the other is a spoofing signal, $\Sigma \rho_{ij}\left(k\right)$ can be derived as
(11)$$\begin{array}{c} \Sigma \rho_{ij}\left(k\right)\approx \Delta {\vec{r}}_{s,i}\left(k\right)\cdot {\vec{l}}_{i}+\Delta {\vec{r}}_{s,j}\left(k\right)\cdot {\vec{l}}_{j}^{p}-\Delta {\vec{r}}_{u}\left(k\right)\cdot \left({\vec{l}}_{i}+{\vec{l}}^{u}\left(k\right)\right)+\Delta \rho_{p,j}\left(k\right)\\ +2c\cdot \delta t_{u}\left(k\right)+\varepsilon_{i}\left(k\right)+\varepsilon_{j}\left(k\right)+\Sigma {\mathrm{Cn}}_{ij} \end{array}.$$

For the two cases, since ${\vec{l}}^{u}\left(k\right)$ will change during the motion of the receiver, $\Delta {\vec{r}}_{u}\left(k\right)\cdot {\vec{l}}^{u}\left(k\right)$ will change with time regardless of whether the receiver moves at a uniform velocity, which means that the elements in $\Sigma {\vec{\rho }}_{ij}$ are nonlinear.

### 2.4. Analysis Summary

Based on all the above analysis, we can draw a conclusion, as shown in Table 1. Case 1 indicates that the two signals are authentic signals. Case 2 indicates that the two signals are spoofing signals. Case 3 indicates that one of the two signals is an authentic signal and the other is a spoofing signal.

## 3. Anti-Spoofing Method

From the analysis in Section 2, it can be seen that when the two signals are spoofing signals, the PRD sequence $\Delta {\vec{\rho }}_{ij}$ is linear, while the PRS sequence $\Sigma {\vec{\rho }}_{ij}$ is nonlinear. In other cases, this characteristic is not satisfied. Accordingly, the detection result $\Delta {\vec{\rho }}_{ij}$ is linear and $\Sigma {\vec{\rho }}_{ij}$ is nonlinear, indicating that the signal i and the signal j are spoofing signals from the same spoofing source. Although no definite conclusion can be obtained from other detection results, all authentic signals and spoofing signals can still be distinguished with traversal detection.

The specific detection process includes the following two steps.

✧Step 1: Randomly select two signals and use the proposed method to detect whether they are both spoofing signals. If yes, proceed to Step 2. If not, we will reselect two signals for detection until two spoofing signals are selected.✧Step 2: For the other signals, except the two spoofing signals, one of them is selected in turn to be jointly detected with any one of the spoofing signals.

For Step 2, on the premise that one signal is a spoofing signal, the proposed method can directly determine whether the other signal is an authentic signal or a spoofing signal. As a result, all authentic signals and spoofing signals can be judged.

The above spoofing detection process is represented with a flowchart, as shown in Figure 1. Next, this section will introduce the detection method of the linearities of PRD sequence $\Delta {\vec{\rho }}_{ij}$ and PRS sequence $\Sigma {\vec{\rho }}_{ij}$ involved in the spoofing detection process.

### 3.1. PRD Sequence Linearity Detection

In the linearity detection of the PRD sequence, two hypotheses for PRD sequence $\Delta {\vec{\rho }}_{ij}$ are as follows:(12)$$\left\{ \begin{array}{l} H_{\Delta ,0}:\Delta {\vec{\rho }}_{ij}\;\mathrm{is}\;\mathrm{nonlinear}\\ H_{\Delta ,1}:\Delta {\vec{\rho }}_{ij}\;\mathrm{is}\;\mathrm{linear} \end{array}\right..$$

For $\Delta {\vec{\rho }}_{ij}$, we can calculate its least square linear regression model:(13)$$\Delta {\overset{⌢}{\rho }}_{ij}\left(k\right)={\overset{⌢}{a}}_{\Delta ,ij}\times k+{\overset{⌢}{b}}_{\Delta ,ij},$$
where ${\overset{⌢}{a}}_{\Delta ,ij}$ and ${\overset{⌢}{b}}_{\Delta ,ij}$ are the estimated model parameters and $\Delta {\overset{⌢}{\rho }}_{ij}\left(k\right)$ represents the estimated value of $\Delta \rho_{ij}\left(k\right)$ obtained with a linear model.

Under the $H_{\Delta ,0}$ hypothesis, $\Delta \rho_{ij}\left(k\right)$ can be expressed as
(14)$$\Delta \rho_{ij}\left(k\right)=g_{\Delta ,ij}\left(k\right)+w_{\Delta ,0}\left(k\right),$$
where $g_{\Delta ,ij}\left(k\right)$ represents a nonlinear equation and $w_{\Delta ,0}\left(k\right)$ is Gaussian white noise. The variance of $w_{\Delta ,0}\left(k\right)$ is $\sigma_{ij}^{2}$:(15)$$\sigma_{ij}^{2}=\sigma_{i}^{2}+\sigma_{j}^{2}.$$

$\sigma_{i}^{2}$ and $\sigma_{j}^{2}$ denote the pseudo-range measurement noise variance of the signal i and the signal j, respectively. In this case, the difference between the actual value and the estimated value $\Delta x_{ij}\left(k\right)$ can be deduced as follows:(16)$$\begin{array}{ll} \Delta x_{ij}\left(k\right) & =\Delta \rho_{ij}\left(k\right)-\Delta {\overset{⌢}{\rho }}_{ij}\left(k\right)\\ & =\left(g_{\Delta ,ij}\left(k\right)+w_{\Delta ,0}\left(k\right)\right)-\left({\overset{⌢}{a}}_{\Delta ,ij}\times k+{\overset{⌢}{b}}_{\Delta ,ij}\right)\\ & =\mu_{\Delta ,ij}\left(k\right)+w_{\Delta ,0}\left(k\right) \end{array}.$$

Therefore, $\Delta x_{ij}\left(k\right)$ obeys the Gaussian distribution with mean value $\mu_{\Delta ,ij}\left(k\right)$ and variance $\sigma_{ij}^{2}$.

Under the $H_{\Delta ,1}$ hypothesis, $\Delta \rho_{ij}\left(k\right)$ can be expressed as
(17)$$\Delta \rho_{ij}\left(k\right)=a_{\Delta ,ij}\times k+b_{\Delta ,ij}+w_{\Delta ,1}\left(k\right),$$
where $a_{\Delta ,ij}$ and $b_{\Delta ,ij}$ are model parameters, $w_{\Delta ,1}\left(k\right)$ is Gaussian white noise, and its variance is also $\sigma_{ij}^{2}$. Thus, the difference between the actual value and the estimated value $\Delta x_{ij}\left(k\right)$ can be deduced as follows:(18)$$\begin{array}{ll} \Delta x_{ij}\left(k\right) & =\Delta \rho_{ij}\left(k\right)-\Delta {\overset{⌢}{\rho }}_{ij}\left(k\right)\\ & =\left(a_{\Delta ,ij}\times k+b_{\Delta ,ij}+w_{\Delta ,1}\left(k\right)\right)-\left({\overset{⌢}{a}}_{\Delta ,ij}\times k+{\overset{⌢}{b}}_{\Delta ,ij}\right)\\ & =\left(a_{\Delta ,ij}-{\overset{⌢}{a}}_{\Delta ,ij}\right)\times k+\left(b_{\Delta ,ij}-{\overset{⌢}{b}}_{\Delta ,ij}\right)+w_{\Delta ,1}\left(k\right)\\ & \approx w_{\Delta ,1}\left(k\right) \end{array}.$$

Compared with $w_{\Delta ,1}\left(k\right)$, $a_{\Delta ,ij}-{\overset{⌢}{a}}_{\Delta ,ij}$ and $b_{\Delta ,ij}-{\overset{⌢}{b}}_{\Delta ,ij}$ are very small and can be ignored. Therefore, $\Delta x_{ij}\left(k\right)$ obeys the Gaussian distribution with zero mean and $\sigma_{ij}^{2}$ variance.

$\Delta {\vec{x}}_{ij}=\left[\begin{array}{ccc} \Delta x_{ij}\left(0\right), & \begin{array}{cc} \cdots & \Delta x_{ij}\left(k\right), \end{array} & \begin{array}{cc} \cdots & \Delta x_{ij}\left(K-1\right) \end{array} \end{array}\right]$ is called the fitting error sequence of the PRD sequence. The effect of linear fitting can be evaluated using the sum of squares of errors (SSE) between actual data and estimated data. The smaller the SSE, the better the linear fitting effect. Based on this, we construct the linearity detection statistic $\Delta T_{ij}$ corresponding to the PRD sequence:(19)$$\Delta T_{ij}=\frac{1}{K}\sum_{k=0}^{K-1}\Delta x_{ij}{\left(k\right)}^{2}=\sum_{k=0}^{K-1}{\left(\frac{\Delta x_{ij}\left(k\right)}{\sqrt{K}}\right)}^{2}.$$

It can be proved that under the hypothesis of $H_{\Delta ,0}$, $\Delta T_{ij}$ obeys the non-central chi-square distribution with the degree of freedom K and the non-central parameter $\lambda_{\Delta ,ij}$. And under the $H_{\Delta ,1}$ hypothesis, $\Delta T_{ij}$ obeys the central chi-square distribution with the degree of freedom K.
(20)$$\left\{ \begin{array}{l} H_{\Delta ,0}:\Delta T_{ij}\sim \chi^{2}\left(K,\lambda_{\Delta ,ij},\frac{\sigma_{ij}^{2}}{K}\right)\\ H_{\Delta ,1}:\Delta T_{ij}\sim \chi^{2}\left(K,0,\frac{\sigma_{ij}^{2}}{K}\right) \end{array}\right..$$

The non-central parameter $\lambda_{\Delta ,ij}$ is as follows:(21)$$\lambda_{\Delta ,ij}=\sum_{k=0}^{K-1}{\left(\frac{\mu_{\Delta ,ij}\left(k\right)}{\sqrt{K}}\right)}^{2}=\frac{1}{K}\sum_{k=0}^{K-1}{\left(\mu_{\Delta ,ij}\left(k\right)\right)}^{2}.$$

When the degree of freedom is large enough, the chi-square distribution can be approximately Gaussian distribution [26]. According to this characteristic of chi-square distribution, the distribution of $\Delta T_{ij}$ under the $H_{\Delta ,0}$ hypothesis and the $H_{\Delta ,1}$ hypothesis can be approximately expressed as
(22)$$\left\{ \begin{array}{l} H_{\Delta ,0}:\Delta T_{ij}\sim N\left(\sigma_{ij}^{2}+\lambda_{\Delta ,ij},\frac{2\sigma_{ij}^{4}}{K}+\frac{4\sigma_{ij}^{2}}{K}\lambda_{\Delta ,ij}\right)\\ H_{\Delta ,1}:\Delta T_{ij}\sim N\left(\sigma_{ij}^{2},\frac{2\sigma_{ij}^{4}}{K}\right) \end{array}\right..$$

Figure 2 shows the probability density function of the linearity detection statistic $\Delta T_{ij}$ under different hypotheses, where the solid lines are the theoretical curves calculated according to the Equation (22), and the star points are the statistical results obtained through 10,000 Monte Carlo random experiments. Figure 2a,b correspond to the $H_{\Delta ,0}$ hypothesis, where their central parameters are $\lambda_{\Delta ,ij}=100$ and $\lambda_{\Delta ,ij}=200$, respectively, while Figure 2c corresponds to the $H_{\Delta ,1}$ hypothesis. It can be found that the simulation results are very close to the theoretical results. Under the $H_{\Delta ,1}$ hypothesis, $\Delta T_{ij}$ is closer to zero, and the smaller the noise variance and the longer the sequence length, the more concentrated the distribution of $\Delta T_{ij}$. Under the $H_{\Delta ,0}$ hypothesis, the mean and variance of $\Delta T_{ij}$ increase with the increase in central parameters.

### 3.2. PRS Sequence Linearity Detection

In the linearity detection of the PRS sequence, two hypotheses for PRS sequence $\Sigma {\vec{\rho }}_{ij}$ are as follows:(23)$$\left\{ \begin{array}{l} H_{\Sigma ,0}:\Sigma {\vec{\rho }}_{ij}\;\mathrm{is}\;\mathrm{linear}\\ H_{\Sigma ,1}:\Sigma {\vec{\rho }}_{ij}\;\mathrm{is}\;\mathrm{nonlinear} \end{array}\right..$$

Similarly, the difference between the actual value and the estimated value obtained with linear fitting $\Sigma {\vec{\rho }}_{ij}$ is represented by $\Sigma x_{ij}\left(k\right)$. Under the $H_{\Sigma ,0}$ hypothesis, $\Sigma x_{ij}\left(k\right)$ obeys the Gaussian distribution with zero mean and $\sigma_{ij}^{2}$ variance. Under the $H_{\Sigma ,1}$ hypothesis, $\Sigma x_{ij}\left(k\right)$ obeys the Gaussian distribution with mean value $\mu_{\Sigma ,ij}\left(k\right)$ and variance $\sigma_{ij}^{2}$. $\Sigma {\vec{x}}_{ij}=\left[\begin{array}{ccc} \Sigma x_{ij}\left(0\right), & \begin{array}{cc} \cdots & \Sigma x_{ij}\left(k\right), \end{array} & \begin{array}{cc} \cdots & \Sigma x_{ij}\left(K-1\right) \end{array} \end{array}\right]$ is called the fitting error sequence of the PRS sequence. Constructing the linearity detection statistic $\Sigma T_{ij}$ corresponding to the PRS sequence:(24)$$\Sigma T_{ij}=\frac{1}{K}\sum_{k=0}^{K-1}\Sigma x_{ij}{\left(k\right)}^{2}=\sum_{k=0}^{K-1}{\left(\frac{\Sigma x_{ij}\left(k\right)}{\sqrt{K}}\right)}^{2}.$$

It can be proved that under the hypothesis of $H_{\Sigma ,0}$, $\Sigma T_{ij}$ obeys the central chi-square distribution with the degree of freedom K. And under the $H_{\Delta ,1}$ hypothesis, $\Sigma T_{ij}$ obeys the non-central chi-square distribution with the degree of freedom K and the non-central parameter $\lambda_{\Sigma ,ij}$.
(25)$$\left\{ \begin{array}{l} H_{\Sigma ,0}:\Sigma T_{ij}\sim \chi^{2}\left(K,0,\frac{\sigma_{ij}^{2}}{K}\right)\\ H_{\Sigma ,1}:\Sigma T_{ij}\sim \chi^{2}\left(K,\lambda_{\Sigma ,ij},\frac{\sigma_{ij}^{2}}{K}\right) \end{array}\right..$$

The non-central parameter $\lambda_{\Sigma ,ij}$ is as follows:(26)$$\lambda_{\Sigma ,ij}=\sum_{k=0}^{K-1}{\left(\frac{\mu_{\Sigma ,ij}\left(k\right)}{\sqrt{K}}\right)}^{2}=\frac{1}{K}\sum_{k=0}^{K-1}{\left(\mu_{\Sigma ,ij}\left(k\right)\right)}^{2}.$$

Combined with the previous analysis, when the degree of freedom is large enough, the chi-square distribution can be approximately Gaussian distribution. Therefore, the distribution of $\Sigma T_{ij}$ under different hypotheses can be derived:(27)$$\left\{ \begin{array}{l} H_{\Sigma ,0}:\Sigma T_{ij}\sim N\left(\sigma_{ij}^{2}\;,\frac{2\sigma_{ij}^{4}}{K}\right)\\ H_{\Sigma ,1}:\Sigma T_{ij}\sim N\left(\sigma_{ij}^{2}+\lambda_{\Sigma ,ij}\;,\frac{2\sigma_{ij}^{4}}{K}+\frac{4\sigma_{ij}^{2}}{K}\lambda_{\Sigma ,ij}\right) \end{array}\right..$$

Figure 3 shows the probability density function of the linearity detection statistic $\Sigma T_{ij}$ under different hypotheses, where the solid lines are the theoretical curves calculated according to the Equation (27), and the star points are the statistical results obtained through 10,000 Monte Carlo random experiments. Figure 3a corresponds to the $H_{\Sigma ,0}$ hypothesis, while Figure 3b,c correspond to the $H_{\Sigma ,1}$ hypothesis, where their central parameters are $\lambda_{\Sigma ,ij}=100$ and $\lambda_{\Sigma ,ij}=200$, respectively. It can be found that the simulation results are very close to the theoretical results. Under the $H_{\Sigma ,0}$ hypothesis, $\Sigma T_{ij}$ is closer to zero, and the smaller the noise variance and the longer the sequence length, the more concentrated the distribution of $\Sigma T_{ij}$. Under the $H_{\Sigma ,0}$ hypothesis, the mean and variance of $\Sigma T_{ij}$ increase with the increase in central parameters.

### 3.3. Detection Performance Analysis

From the analysis in Section 2, it can be seen that the noise contained with the PRD $\Delta \rho_{ij}\left(k\right)$ is $\varepsilon_{i}\left(k\right)-\varepsilon_{j}\left(k\right)$, and the noise contained with PRS $\Sigma \rho_{ij}\left(k\right)$ is $\varepsilon_{i}\left(k\right)+\varepsilon_{j}\left(k\right)$. $\varepsilon_{i}\left(k\right)$ and $\varepsilon_{j}\left(k\right)$ are not correlated. Therefore, the variance of $\varepsilon_{i}\left(k\right)-\varepsilon_{j}\left(k\right)$ and $\varepsilon_{i}\left(k\right)+\varepsilon_{j}\left(k\right)$ is $\sigma_{ij}^{2}=\sigma_{i}^{2}+\sigma_{j}^{2}$. The correlation coefficient $\eta_{ij}$ between them can be derived as
(28)$$\eta_{ij}=\frac{\mathrm{cov}\left(\varepsilon_{i}\left(k\right)-\varepsilon_{j}\left(k\right),\varepsilon_{i}\left(k\right)+\varepsilon_{j}\left(k\right)\right)}{\sqrt{\mathrm{var}\left(\varepsilon_{i}\left(k\right)-\varepsilon_{j}\left(k\right)\right)\cdot \mathrm{var}\left(\varepsilon_{i}\left(k\right)+\varepsilon_{j}\left(k\right)\right)}}=\frac{\sigma_{i}^{2}-\sigma_{j}^{2}}{\sigma_{i}^{2}+\sigma_{j}^{2}}.$$

When the pseudo-range measurement noise variances $\sigma_{i}^{2}$ and $\sigma_{j}^{2}$ are equal, the correlation coefficient is $\eta_{ij}=0$, which means that the PRD sequence $\Delta {\vec{\rho }}_{ij}$ is not correlated with the noise contained in the PRS sequence $\Sigma {\vec{\rho }}_{ij}$. When $\sigma_{i}^{2}\neq \sigma_{j}^{2}$, $\Delta {\vec{\rho }}_{ij}$ is correlated with the noise contained in $\Sigma {\vec{\rho }}_{ij}$, and the greater the difference between the two noise variances, the higher the correlation. Considering that the spoofing signals broadcast by the same spoofing source have similar power, to simplify the problem, it is assumed that the noises contained in $\Delta {\vec{\rho }}_{ij}$ and $\Sigma {\vec{\rho }}_{ij}$ are not correlated. That is to say, the detection of PRD sequence linearity is independent of the detection of PRS sequence linearity.

For the PRD sequence and PRS sequence linearity detection, since the noise variance is $\sigma_{ij}^{2}$ and the detection length is K, the same detection threshold γ can be set. In the linearity detection of the PRD sequence, the $H_{\Delta ,0}$ hypothesis is considered to be valid when $\Delta T_{ij}>\gamma$, and the $H_{\Delta ,1}$ hypothesis is considered to be valid when $\Delta T_{ij}\leq \gamma$. Combined with the Equation (22), the false alarm probability $P_{\Delta ,\mathrm{fa}}$ and the detection probability $P_{\Delta ,d}$ can be derived as follows:
(29)$$\left\{ \begin{array}{l} P_{\Delta ,\mathrm{fa}}=P\left(\Delta T_{ij}\leq \gamma |H_{\Delta ,0}\right)\approx 1-Q\left(\frac{\gamma -\left(\sigma_{ij}^{2}+\lambda_{\Delta ,ij}\right)}{\sqrt{\frac{2\sigma_{ij}^{4}}{K}+\frac{4\sigma_{ij}^{2}}{K}\lambda_{\Delta ,ij}}}\right)\\ P_{\Delta ,d}=P\left(\Delta T_{ij}\leq \gamma |H_{\Delta ,1}\right)\approx 1-Q\left(\frac{\gamma -\sigma_{ij}^{2}}{\sqrt{\frac{2\sigma_{ij}^{4}}{K}}}\right) \end{array}\right..$$
where Q(⋅) is the right tail probability function. In the linearity detection of the PRS sequence, the $H_{\Sigma ,0}$ hypothesis is considered to be valid when $\Sigma T_{ij}\leq \gamma$, and the $H_{\Sigma ,1}$ hypothesis is considered to be valid when $\Sigma T_{ij}>\gamma$. Combined with the Equation (27), the false alarm probability $P_{\Sigma ,\mathrm{fa}}$ and the detection probability $P_{\Sigma ,d}$ can be derived as follows:(30)$$\left\{ \begin{array}{l} P_{\Sigma ,\mathrm{fa}}=P\left(\Sigma T_{ij}>\gamma |H_{\Sigma ,0}\right)\approx Q\left(\frac{\gamma -\sigma_{ij}^{2}}{\sigma_{ij}^{2}\sqrt{\frac{2}{K}}}\right)\\ P_{\Sigma ,d}=P\left(\Sigma T_{ij}>\gamma |H_{\Sigma ,1}\right)\approx Q\left(\frac{\gamma -\left(\sigma_{ij}^{2}+\lambda_{\Sigma ,ij}\right)}{\sqrt{\frac{2\sigma_{ij}^{4}}{K}+\frac{4\sigma_{ij}^{2}}{K}\lambda_{\Sigma ,ij}}}\right) \end{array}\right..$$

For the method proposed in this paper, the false alarm probability $P_{\mathrm{fa}}$ and detection probability $P_{d}$ of joint detection can be expressed as
(31)$$\{\begin{array}{l} P_{\mathrm{fa}}=P\left(\Delta T_{ij}\leq \gamma \;\mathrm{and}\;\Sigma T_{ij}>\gamma |\left(H_{\Delta ,0}\;\mathrm{or}\;H_{\Sigma ,0}\right)\right)\\ P_{d}=P\left(\Delta T_{ij}\leq \gamma \;\mathrm{and}\;\Sigma T_{ij}>\gamma |\left(H_{\Delta ,1}\;\mathrm{and}\;H_{\Sigma ,1}\right)\right) \end{array}.$$

Since it is assumed that the PRD sequence linearity detection is independent of the PRS sequence linearity detection, $P_{\mathrm{fa}}$ and $P_{d}$ can be further derived:(32)$$\begin{array}{ll} P_{\mathrm{fa}} & =P\left(\Delta T_{ij}\leq \gamma |H_{\Delta ,0}\right)\cdot P\left(\Sigma T_{ij}>\gamma |H_{\Sigma ,0}\right)+P\left(\Delta T_{ij}\leq \gamma |H_{\Delta ,1}\right)\cdot P\left(\Sigma T_{ij}>\gamma |H_{\Sigma ,0}\right)\\ & +P\left(\Delta T_{ij}\leq \gamma |H_{\Delta ,0}\right)\cdot P\left(\Sigma T_{ij}>\gamma |H_{\Sigma ,1}\right)\\ & =P_{\Delta ,\mathrm{fa}}\cdot P_{\Sigma ,\mathrm{fa}}+P_{\Delta ,d}\cdot P_{\Sigma ,\mathrm{fa}}+P_{\Delta ,\mathrm{fa}}\cdot P_{\Sigma ,d} \end{array},$$
(33)$$P_{d}=P\left(\Delta T_{ij}\leq \gamma |H_{\Delta ,1}\right)\cdot P\left(\Sigma T_{ij}>\gamma |H_{\Sigma ,1}\right)=P_{\Delta ,d}\cdot P_{\Sigma ,d}.$$

The false alarm probability $P_{\mathrm{fa}}$ and detection probability $P_{d}$ of joint detection corresponding to the detection threshold γ in different scenes are shown in Figure 4. Compared with the Figure 4a,b, the increase in noise variance makes $P_{\mathrm{fa}}$ higher and $P_{d}$ lower. Compared with the Figure 4a,c, the reduction in sequence length makes $P_{\mathrm{fa}}$ higher and $P_{d}$ lower. Compared with the Figure 4a,d, the decrease in non-central parameter $\lambda_{\Delta ,ij}$ makes $P_{\mathrm{fa}}$ increase, and $P_{d}$ remains unchanged. Compared with the Figure 4a,e, the decrease in non-central parameter $\lambda_{\Sigma ,ij}$ reduces $P_{\mathrm{fa}}$ and $P_{d}$.

The lower the false alarm probability and the higher the detection probability, the better the detection performance. Furthermore, we count the difference $P_{d}-P_{\mathrm{fa}}$ between the detection probability and the false alarm probability corresponding to the detection threshold γ in different scenes, as shown in Figure 5. It is not difficult to find from the figure that the smaller the noise variance, the longer the sequence length, and the larger the non-central parameters $\lambda_{\Delta ,ij}$ and $\lambda_{\Sigma ,ij}$, the better the detection performance.

## 4. Simulation Results and Analysis

In this section, the feasibility of the method will be verified with simulation tests. Simulation scenes are divided into the uniform motion scene and the circular motion scene, as shown in Figure 6. The authentic signals and spoofing signals in the two scenes are generated with the GNSS signal source simulator, sampled and stored in signal memory, and then processed with the GNSS software receiver.

The number of test signals is 8, in which the PRNs of authentic signals are 2, 11, 14, and 21, and the PRNs of spoofing signals are 5, 10, 16, and 33. The standard deviation of the pseudo-range measurement error of all signals is set to 3 m. This assumption implies that the noise variance of PRD and PRS for any two signals is 18. The receiver tracks the collected signals and obtains the measured pseudo-range sequence with the sequence length K=120 and the time interval Δt=0.1 s. The difference between two measurement pseudo-range sequences is selected to form a PRD sequence, and the sum forms a PRS sequence. Each PRD sequence and PRS sequence is linearly fitted to obtain fitting error sequences, and the detection statistics are obtained by statistics. The detection threshold γ=30. For this threshold, the probability that the sequence is linear but the detection result is nonlinear will be less than 0.001%.

### 4.1. Uniform Motion Scene Test

In the uniform motion scene test, the distance between the spoofing source and the target receiver is 1000 m at 0 time, and the speed of the receiver is 40 m/s. The fitting error sequences $\Delta {\vec{x}}_{ij}$ and $\Sigma {\vec{x}}_{ij}$ correspond to the PRD sequence and the PRS sequence, and the linearity detection statistics $\Delta T_{ij}$ and $\Sigma T_{ij}$ obtained by statistics are shown in Figure 7. In the case of two authentic signals, as shown in Figure 7a, both $\Delta T_{ij}$ and $\Sigma T_{ij}$ are below the threshold. For the case of one authentic signal and one spoofing signal, as shown in Figure 7c, both $\Delta T_{ij}$ and $\Sigma T_{ij}$ are above the threshold. For the case of two spoofing signals, as shown in Figure 7b, $\Delta T_{ij}\leq \gamma$ and $\Sigma T_{ij}>\gamma$ are satisfied. The simulation results are consistent with the analysis results, which show that the proposed method can detect spoofing signals in uniform motion scenes.

### 4.2. Circular Motion Scene Test

In the circular motion scene test, the distance between the spoofing source and the target receiver is 1000 m at 0 time, the speed of the receiver is 40 m/s, and the circumference radius is 300 m. The fitting error sequences $\Delta {\vec{x}}_{ij}$ and $\Sigma {\vec{x}}_{ij}$ correspond to the PRD sequence and the PRS sequence, and the linearity detection statistics $\Delta T_{ij}$ and $\Sigma T_{ij}$ obtained with the test are shown in Figure 8. In the case of two authentic signals, as shown in Figure 8a, both $\Delta T_{ij}$ and $\Sigma T_{ij}$ are above the threshold. For the case of one authentic signal and one spoofing signal, as shown in Figure 8c, both $\Delta T_{ij}$ and $\Sigma T_{ij}$ are above the threshold. For the case of two spoofing signals, as shown in Figure 8b, $\Delta T_{ij}\leq \gamma$ and $\Sigma T_{ij}>\gamma$ are satisfied. The simulation results are consistent with the analysis results, which show that the proposed method can detect spoofing signals in variable motion scenes.

## 5. Conclusions

Aiming at single spoofing interference, this paper proposes an anti-spoofing method based on the joint monitoring of the PRD sequence and the PRS sequence linearity, which changes the spoofing detection problem into the sequence linearity detection problem. The detection statistics are constructed based on the SSE of linear fitting of the PRD sequence and PRS sequence. When the sequence length is large enough, the detection statistics approximately obey the Gaussian distribution. The influence of noise variance, sequence length, and non-central parameters on the detection performance of the method is analyzed. It can be concluded that the smaller the noise variance, the longer the sequence length, and the larger the non-central parameters, the better the detection performance. In addition, the feasibility of the method is verified by simulation tests. Simulation results show that this method can detect spoofing interference and distinguish authentic signals from spoofing signals in both uniform motion scenes and variable motion scenes. This is the premise of improving the security and availability of GNSS in the presence of spoofing interference.

The proposed method is low complexity because it only requires pseudo-range information and does not need to modify the baseband processing of the receiver. Nevertheless, the precondition for the effectiveness of the method is that the spoofing source is stationary, and the additional component of the spoofing source to any signal is linearly varying. In addition, this paper lacks an exploration of the influence of receiver motion on non-central parameters, which makes it impossible to establish a direct relationship between receiver motion and detection performance. Our future work will focus on these two aspects to improve.

## Figures and Tables

**Figure 1 sensors-23-08418-f001:**
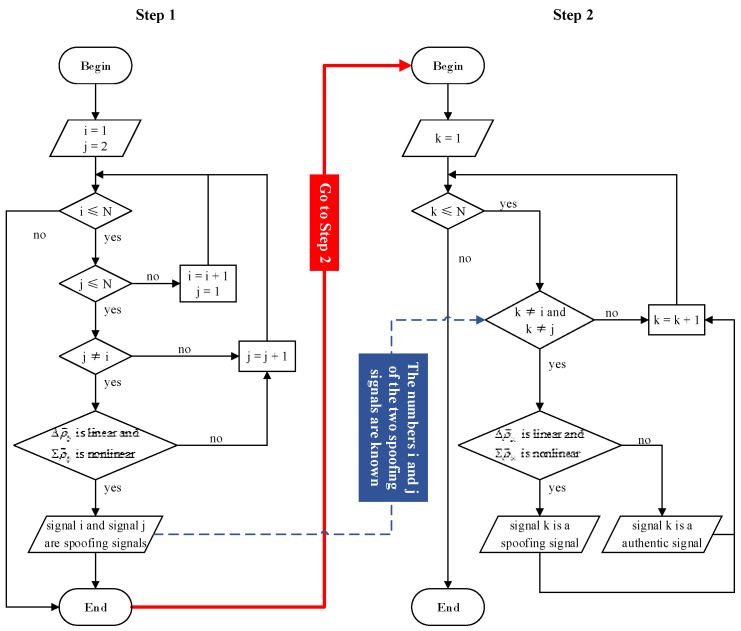
Flowchart corresponding to spoofing detection process.

**Figure 2 sensors-23-08418-f002:**
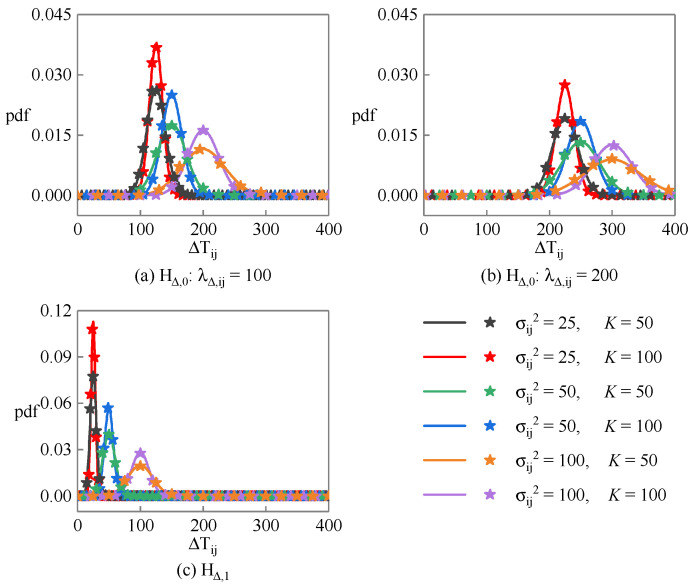
Probability density function of the linearity detection statistic corresponding to PRD sequence under different hypotheses.

**Figure 3 sensors-23-08418-f003:**
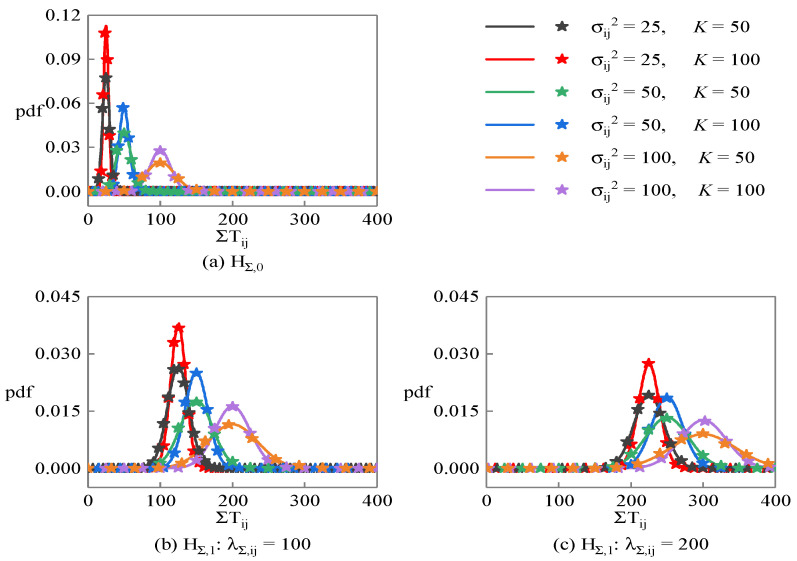
Probability density function of the linearity detection statistic corresponding to PRS sequence under different hypotheses.

**Figure 4 sensors-23-08418-f004:**
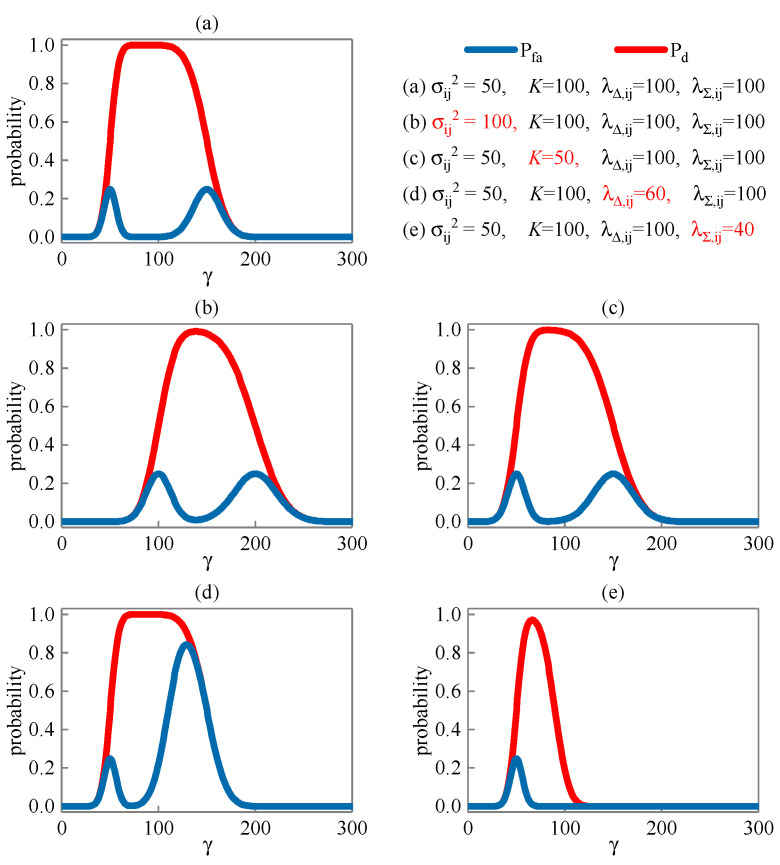
False alarm probability and detection probability corresponding to detection threshold.

**Figure 5 sensors-23-08418-f005:**
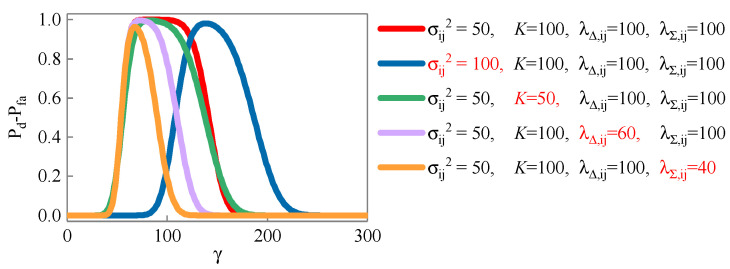
Difference between detection probability and false alarm probability corresponding to detection threshold.

**Figure 6 sensors-23-08418-f006:**
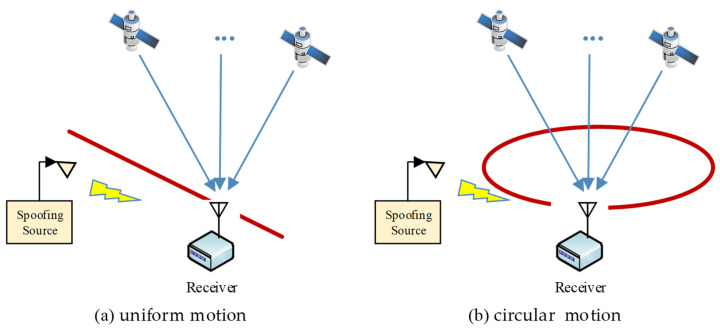
Two simulation scenes.

**Figure 7 sensors-23-08418-f007:**
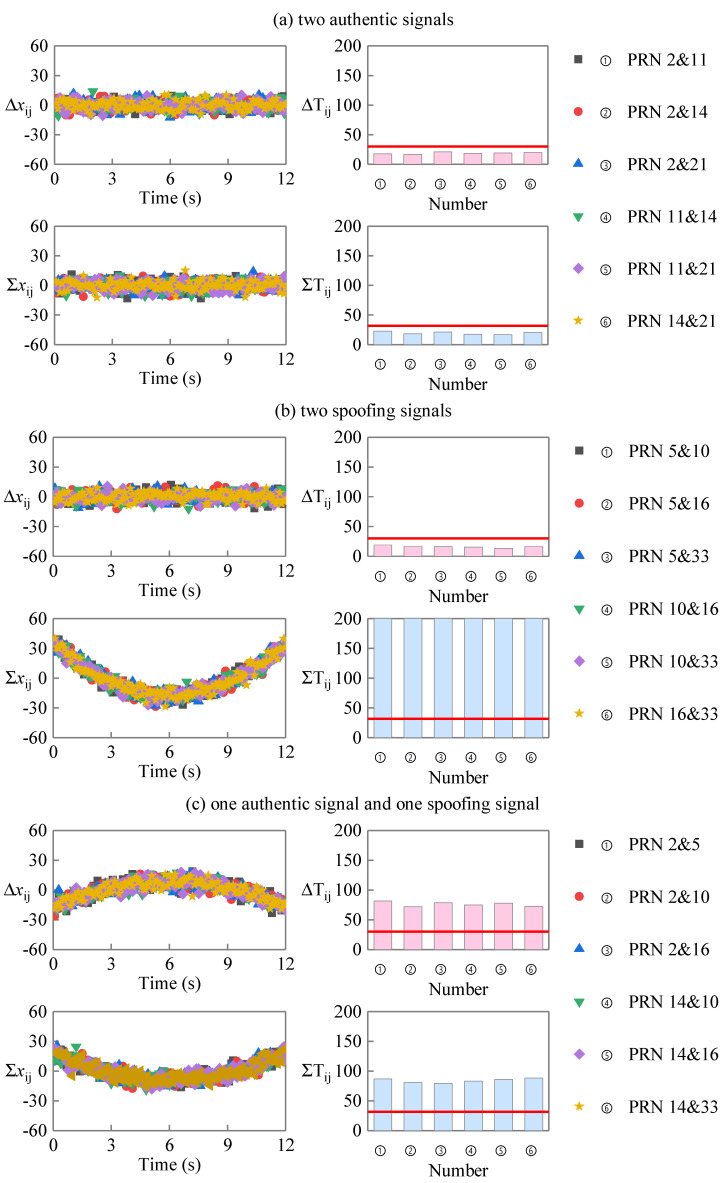
Fitting error sequences and detection statistics in the uniform motion scene.

**Figure 8 sensors-23-08418-f008:**
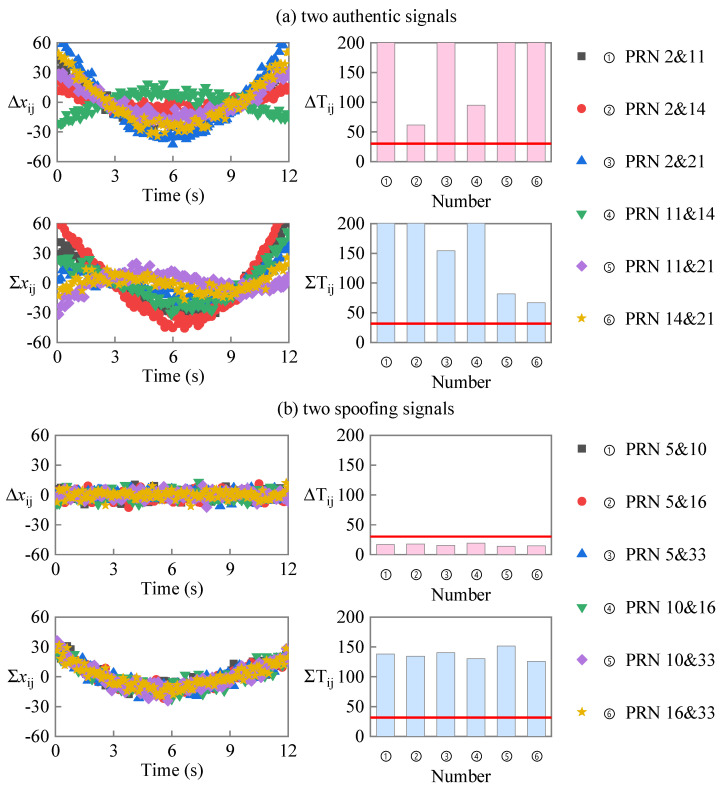
Fitting error sequences and detection statistics in the circular motion scene.

**Table 1 sensors-23-08418-t001:** Linearity analysis results of PRD sequence and PRS sequence.

	Case 1		Case 2		Case 3	
	Constant	Variational	Constant	Variational	Constant	Variational
$\Delta {\vec{\rho }}_{ij}$	linear	nonlinear	linear	linear	nonlinear	nonlinear
$\Sigma {\vec{\rho }}_{ij}$	linear	nonlinear	nonlinear	nonlinear	nonlinear	nonlinear

## Data Availability

Not applicable.

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
