# Peer review of "Anti-Spoofing Method for Improving GNSS Security by Jointly Monitoring Pseudo-Range Difference and Pseudo-Range Sum Sequence Linearity"

_sensors, 2023, doi:10.3390/s23208418_

Round 1
Reviewer 1 Report
In the manuscript titled ‘Anti-Spoofing Method for Improving GNSS Security by Jointly Monitoring Pseudo-Range Difference and Pseudo-Range Sum Sequence Linearity’, the author presents an anti-spoofing method by monitoring statistical characteristics of the pseudorange difference and pseudorange sum sequences from two satellites. This method is validated through simulation tests and discusses several factors affecting its detection performance. However, several questions arise regarding the design of the simulation tests and the spoofing detection algorithm.
Q1: The author has conducted a comprehensive analysis of the linearity of the proposed detection statistics, both theoretically and empirically. However, it might be beneficial to establish a clearer relationship between linearity detection and spoofing detection for each satellite. For example, an illustration of the detection algorithm.
Q2: The proposed linearity detector is based on any two different satellites, and it addresses three possible cases: two authentic satellites, one authentic and one spoofed satellite, and two spoofed satellites. The author's conclusion is that the proposed method effectively distinguishes between authentic signals and spoofing signals. However, in the scenario involving one authentic and one spoofed satellite, it remains unclear how this method distinguishes the spoofed satellite. There is a lack of evaluation for detecting the absence or presence of spoofing for each satellite.
Q3: Chapter 4.2 illustrates the detection performance in a circular motion test. However, the detection statistics exceed the threshold in both cases: when there are two authentic signals and when there's one authentic and one spoofing signal. How can we effectively differentiate between these two scenarios?
Q4: In Row 315, the sequence length of the detector is 120, with a time interval of 0.1s. Does this imply that this detector provides an output decision every 12s? If it is true, whether a detection time of 12s is too long for a real spoofing attack application? Additionally, does a faster receiver pseudo-range update rate benefit your proposed detector?
Q5: row-319, the detection threshold is set to 30. Are there any reasons for this setting and what is the corresponding false alarm rate?
Q6: The author analyzed the influence of noise variance, sequence length, and non-central parameters theoretically. However, in Chapter 4, only the sequence length is addressed among the signal-related parameters during the simulation.
Q7: In the chapter on simulation tests, the speed of the receiver is set to 40m/s (144km/h), a value similar to the highway car speeds in practical applications. Whether the proposed method remain effective in low-speed applications? As the receiver's velocity has a substantial correlation with the proposed detection statistics, it may be valuable to explore whether variations in the magnitude and rate of change of the receiver's velocity might impact the anti-spoofing performance.
Moderate editing of English language required
Reviewer 2 Report
Dear Authors,
I have reviewed your manuscript titled "Anti-Spoofing Method for Improving GNSS Security by Jointly Monitoring Pseudo-Range Difference and Pseudo-Range Sum Sequence Linearity." The topic is timely and relevant, and the methodology presented is innovative. Here are some constructive comments:
-
Clarity and Depth: The paper provides a clear understanding of the spoofing interference problem and the proposed solution. However, it would be beneficial to delve deeper into the practical implications of the proposed method, especially in real-world scenarios.
-
Comparison with Existing Methods: While the paper mentions other methods based on multiple antennas or receivers, a more comprehensive comparison with existing anti-spoofing techniques would enhance the paper's value.
-
Limitations: Every method has its limitations. It would be beneficial for readers if the potential limitations or challenges of the proposed method were discussed.
-
Future Work: While the paper touches upon future considerations, elaborating on potential extensions or improvements to the current method would be insightful.
-
Figures and Visuals: Including more visual representations, such as flowcharts or diagrams, could help readers grasp the methodology more effectively.
-
Simulation Results: It would be helpful to provide more details about the simulation setup, parameters used, and potential real-world data that could validate the findings.
In conclusion, the paper offers a significant contribution to the field of GNSS security. With some refinements and further elaboration on certain aspects, it has the potential to be a pivotal piece in the domain.
Warm regards
Reviewer 3 Report
The authors present an original technique to detect spoofing signals from the linearity of pseudorange differences and sums during motion. The manuscript is very well written, the methodology and mathematical derivations are presented with great care and rigor and the theory is backed by simulation. The proposed method is rather theoretical though, as the requirement that the receiver is driven by a stable clock (line 117) is likely to significantly limit its applicability in real-life conditions.
I only have minor comments:
Line 123: “under the assumption that the position of spoofing source is unchanged”. I’d also stress that the spoofed position must also be static for the technique to work.
Line 239,240: the “Delta” subscript should be a “Sigma” subscript
Line 262: what are the “D” and “P” symbols?
Line 263: it is first said that \epsilon_i-\epsilon_j is not correlated with \epsilon_i+\epsilon_j, while the discussion after equation (28) indicates that they can be correlated depending on the noise variance difference. This should be clarified.
Round 2
Reviewer 1 Report
The authors answered all the comments. The authors further illustrated that the proposed method is effective in a specific spoofing attack scenario. The advantages and drawbacks are well illustrated. However, Figure 1 is not shown properly and I believe more texts should be included in the figure.
Please proofread the manuscript carefully.
Author Response
Dear Reviewer,
We are very glad that our previous reply has been accepted by you, and we wish to express our sincere thanks for you again. Figure 1 has been corrected and some text has been added. The modification position is on line 215. In addition, we have made full checks and partial revisions to the English language. The changes have been marked in the revised paper.
Sincerely,
Xinran Zhang, Taotao Liang, Jie Tian, Junwei Wu, Chun Yang, Maolin Chen*
8, Oct, 2023